# Thermophilic PHP Protein Tyrosine Phosphatases (Cap8C and Wzb) from Mesophilic Bacteria

**DOI:** 10.3390/ijms25021262

**Published:** 2024-01-19

**Authors:** Adepeju Aberuagba, Enoch B. Joel, Adebayo J. Bello, Adedoyin Igunnu, Sylvia O. Malomo, Femi J. Olorunniji

**Affiliations:** 1School of Pharmacy & Biomolecular Sciences, Liverpool John Moores University, Byrom Street, Liverpool L3 3AF, UK; adepaberuagba@gmail.com (A.A.); enjoebest@yahoo.com (E.B.J.); a.j.bello@ljmu.ac.uk (A.J.B.); 2Department of Biochemistry, Faculty of Basic Medical Sciences, University of Jos, Jos 930003, Nigeria; 3Department of Biochemistry, Faculty of Life Sciences, University of Ilorin, Ilorin 234031, Nigeria; doyinigunnu@yahoo.com (A.I.); somalomo@unilorin.edu.ng (S.O.M.)

**Keywords:** protein tyrosine phosphatases, polymerase and histidinol phosphatases, Cap8C, Wzb, metal ion activation, thermophilic enzymes

## Abstract

Protein tyrosine phosphatases (PTPs) of the polymerase and histidinol phosphatase (PHP) superfamily with characteristic phosphatase activity dependent on divalent metal ions are found in many Gram-positive bacteria. Although members of this family are co-purified with metal ions, they still require the exogenous supply of metal ions for full activation. However, the specific roles these metal ions play during catalysis are yet to be well understood. Here, we report the metal ion requirement for phosphatase activities of *S. aureus* Cap8C and *L. rhamnosus* Wzb. AlphaFold-predicted structures of the two PTPs suggest that they are members of the PHP family. Like other PHP phosphatases, the two enzymes have a catalytic preference for Mn^2+^, Co^2+^ and Ni^2+^ ions. Cap8C and Wzb show an unusual thermophilic property with optimum activities over 75 °C. Consistent with this model, the activity–temperature profiles of the two enzymes are dependent on the divalent metal ion activating the enzyme.

## 1. Introduction

Protein tyrosine phosphorylation and dephosphorylation exist in both prokaryotes and eukaryotes playing critical roles in pigment production, biofilm formation, biosynthesis of capsular/extracellular polysaccharides, control of cell growth and cell differentiation, along with other biological functions [1,2]. In these processes, protein tyrosine phosphorylation and dephosphorylation are characterised by the opposing actions of protein tyrosine kinases and protein tyrosine phosphatases, respectively [2].

Protein tyrosine phosphatases (PTPs EC 3.1.3.48) catalyse the removal of phosphate groups from phosphorylated tyrosine residues on proteins [3]. They are classified as classical PTPs, dual-specificity PTPs (DSPs), low-molecular-weight phosphatase (LMPTP) and pTyr-specific phosphatases [4]. Recently, numerous manganese-dependent prokaryotic O-phosphatases have also been characterised and classified based on amino acid sequences into phosphoprotein phosphatase (PPP), Mg^2+^/Mn^2+^-dependent protein phosphatase (PPM) and polymerase and histidinol phosphatase (PHP) superfamilies [5]. Protein tyrosine phosphatases of the PHP superfamily are commonly found in Gram-positive bacteria with catalytic mechanisms dependent on metals [6]. Except for PHP family members, protein tyrosine phosphatases generally share a signature catalytic motif C–X_5_–R–(S/T) [6]. Sequence alignment (Figure 1) shows that the C–X_5_–R–(S/T) motif is absent in PTPs of the PHP superfamily [6,7].

Histidine residues are proposed to be involved in divalent metal ion binding at the catalytic site, but very few site-directed mutagenesis studies support this claim [7]. Divalent metal ions such as manganese, cobalt and copper have been widely reported to activate members of the PHP superfamily while others like magnesium, calcium and nickel are reported to have little impact on phosphatase activity [6,7]. While the histidine residues of the four conserved motifs of the PHP family [8] are proposed to be responsible for divalent metal ion binding, the aspartate residue coordinates the catalytic site [7]. Site-directed mutagenesis studies have revealed that the inactivation of the aspartate or histidine residues of motif IV resulted in the loss of activity of *S. pneumoniae* Cps2B [9]. The finding implied that metal ion binding was necessary for PHP phosphoesterase activity and histidine residues were crucial for the binding. Like typical PHP family members, the full activation of *L. rhamnosus* Wzb is dependent on metal ions [7].

Even though *L. rhamnosus* Wzb originates from a mesophilic organism, the enzyme possesses potential thermophilic properties in the presence of metal ions and may be a potential thermozyme with industrial benefits [7]. The presence of metal ions has been demonstrated to assist some enzymes in maintaining stability at temperatures that are otherwise lethal in the absence of such metal ions [10]. In view of this, it is important to study how different metal ions modify the phosphatase activities and thermophilic potentials of both *S. aureus* Cap8C and *L. rhamnosus* Wzb to be able to work out the possible roles of these metal ions.

The unusual thermophilic property of *L. rhamnosus* Wzb had earlier been reported by LaPointe et al. [7], where the phosphatase activity of the enzyme was observed to peak at 75 °C following a 30 min pre-incubation at the same temperature. Interestingly, the residual activity of Wzb at 50 °C was reported to have declined to 10% after 30 min pre-incubation [7]. These properties revealed that Wzb from a mesophilic source might have interesting thermophilic potentials that can be further investigated. The unusual thermophilic property of Wzb reported by LaPointe et al. [7] was the first among members of the PHP phosphatase family.

In this study, we sought to investigate if this feature can be found in other phosphatases from the same family. Hence, we chose Cap8C, a PHP phosphatase from *S. aureus*, another mesophilic organism. Although the catalytic activities of *S. aureus* Cap8C are yet to be reported in the literature, comparative sequence analyses (Figure 1) show that the phosphatase contained similar highly conserved motifs of the PHP domain as found in *Streptococcus pneumoniae* CpsB (the first characterised member of the PHP family) and *L. rhamnosus* Wzb [7]. Since there are some differences in amino acid sequences of the PHP PTP family members, especially in the conserved motif regions, we chose Cap8C since it shows some sequence divergence from Wzb in these regions. In addition, we wanted to look at another PHP PTP that has not been reported in the literature. We hope to achieve a clearer understanding of the metal ions that activate these enzymes and to see whether the thermophilic property is a common theme in the protein family.

Although the catalytic mechanism of PHP family members is not fully characterised, it appears to rely chiefly on signature sequence motif 4, which consists of five histidine residues and a single aspartic acid residue [11]. Mutations of conserved histidine and aspartic acid residues in the signature motif 4 greatly impair phosphatase activity and the formation of cell capsules, as observed for *S. pneumonia* CpsB [11].

Some structural information on the active site of PTPs of the PHP family reveals that they are co-purified with metal ions bound to the catalytic site but still require an unusual complex mixture of metal ions for their full activation [7,9,12]. LaPointe et al. reported that several divalent metal ions activated *L. rhamnosus* Wzb [7]. Even though these metal ions seem important for enzyme activation, the specific roles they play in catalysis are yet to be fully understood, although some structural roles have been suggested [12,13]. It is also unclear if all the metal ions are needed for activity.

Members of the polymerase and histidinol phosphatase family of proteins have active sites characterised by a trinuclear metal centre and distorted TIM-barrel protein fold [13]. The active site architecture of some members of the PHP family such as CpsB from *S. pneumonia*, *Bacillus subtilis* YwqE and Lactococcus lactis HPP has been determined [12,13,14]. The catalytic structure of PTP of the PHP family can be described with reference to Cps4B from *S. pneumonia*. The structure consists of two three-stranded parallel β-sheets with a ring of α-helices on the outside of the structure [12]. Three metal ions are found bounded adjacent to one another between the β-sheets with three water molecules serving as ligands; the fold can be described as a distorted TIM barrel (Figure 2) [8,12].

Enzymes in the PHP family are co-purified with metal ions bound to the active site; Cps4B is co-purified with three metal ions bound to the active site [12]. However, they still require an exogenous supply of metal ions (particularly divalent metal ions) for full activation [8]. Kim et al. reported that metal analyses of recombinant YwqE and CpsB by inductively coupled plasma atomic emission indicated that they contain mostly iron (Fe) and magnesium (Mg), but not other metal ions such as nickel (Ni^2+^), copper (Cu^2+^), zinc (Zn^2+^), manganese (Mn^2+^) and cobalt (Co^2+^) [13]. Kim et al. also modelled M1, M2 and M3 of YwqE and CpsB as iron (Fe) and magnesium (Mg) ions [13], which is different from what was reported by Hagelueken et al., who modelled M1, M2 and M3 for CpsB as manganese (Mn^2+^) ions [12].

## 2. Results

### 2.1. AlphaFold2-Predicted Structures of Cap8C and Wzb Show the Polymerase and Histidinol Phosphatase (PHP) Domain

The characteristic conserved PHP domains highlighted in the sequence alignment show that Cap8C and Wzb are members of the PHP family (Figure 1). The structures of Cap8C and Wzb, as predicted by AlphaFold2 [15,16], were downloaded from the AlphaFold Protein Structure Database (https://alphafold.ebi.ac.uk/, accessed on 13 January 2024) and visualised using PyMOL (https://pymol.org/2/, accessed on 13 January 2024). Both structures reflect the typical polymerase and histidinol phosphatase (PHP) domain. The PHP domain is characterised by four conserved sequence motifs that contain invariant histidine and aspartate residues that function in coordinating metal ions [17]. The motif has a signature distorted (α/β)7-barrel fold (Figure 1). This unique fold is similar to the TIM-barrel fold found in metallohydrolases. The PHP-like distorted (α/β)7 barrel is also found in the protein YcdX from *E. coli*, which has been shown to display phosphatase activity to p-nitrophenylphosphate (pNPP) [18]. Interestingly, the (α/β)7 barrel has a trinuclear Zn^2+^ metal-binding site on the C-terminal side of the barrel.

### 2.2. Metal Ion Activation of Cap8C and Wzb Activities

Analysis of the AlphaFold-predicted structures of Cap8C and Wzb along with evidence from biochemical characterisation of related PTPs [7] suggest that the two proteins belong to the PHP phosphatase family. The PHP PTPs typically prefer Mn^2+^ as the cofactor, and studies of Wzb showed that it is activated by a wide range of divalent metal ions [7]. To determine the range of Mn^2+^ concentration required for optimal activities of the two enzymes, we measured their activities in the presence of increasing concentrations of the metal ion.

As shown in Figure 3, we tested a range of concentrations from 0 to 1000 µM. The two enzymes show a correlation between phosphatase activity and Mn^2+^ concentration. While Wzb shows a significant increase in activity from 250 to 1000 µM Mn^2+^, there was only a marginal change in Cap8C as the cofactor concentration was serially doubled to 500 and 1000 µM. Based on this result, we chose to test the metal ion specificities of the two enzymes using the potential cofactors at 1000 µM (1 mM) concentration.

To confirm which metal ions activated Cap8C, we tested the effect of selected divalent metal cofactors (Co^2+^, Ni^2+^, Mn^2+^, Mg^2+^ and Ca^2+^) on the phosphatase activities of Cap8C and Wzb. *Staphylococcus aureus* Cap8C and *Lactobacillus rhamnosus* Wzb proteins were established to have phosphatase activity using pNPP as substrate. The protein content of samples of the purified enzyme solutions of both Cap8C and Wzb were determined as 0.106 mg/mL (3.61 µmol/mL) and 0.099 mg/mL (3.43 µmol/mL), respectively. From these solutions, 2.5 µM solutions were prepared which were then used for subsequent experiments.

As shown in Figure 4, Cap8C and Wzb showed very low activities (barely detectable) when the reaction buffer was not supplemented with any metal ions. Three of the divalent cations studied (Co^2+^, Ni^2+^, Mn^2+^) showed significant activation of the activities of both enzymes, while Mg^2+^, Ca^2+^, Cu^2+^, Zn^2+^ and Fe^2+^ did not have any noticeable effects on the activities of Cap8C and Wzb. This confirms that the two PTPs behaved like the PHP group of protein tyrosine phosphatases. Co^2+^ (1.0 mM) had a significant activating effect on Cap8C and Wzb activities, yielding a 17- and 29-fold increase in activity, respectively. In the presence of Ni^2+^ (1.0 mM), the activity of Cap8C increased 12-fold, while Wzb had a 4-fold increase in activity. Wzb showed a 24-fold increase in activity in the presence of 1.0 mM Mn^2+^, while Cap8C showed a 7-fold increase in activity.

In summary, among the divalent cations that we studied, Co^2+^, Ni^2+^ and Mn^2+^ had significant activating effects on Cap8C and Wzb in the order Co^2+^ (17-fold) > Ni^2+^ (12-fold) > Mn^2+^ (7-fold) and Co^2+^ (29-fold) > Mn^2+^ (24-fold) > Ni^2+^ (4-fold), respectively.

### 2.3. Concentration-Dependent Activation of Cap8c and Wzb by Co^2+^, Ni^2+^ and Mn^2+^ Ions

Having established that Cap8C and Wzb are activated to different extents by Co^2+^, Ni^2+^ and Mn^2+^ ions, we determined the effect of the concentration of each metal ion on the activities of the two enzymes. We added increasing concentrations (0.25, 0.5, 1.0 and 2.0 mM) of the three divalent cofactors to the reaction buffer and measured the activity of the two enzymes.

Cap8C: As shown in Figure 5, all three metal ions caused a concentration-dependent and significant (*p* < 0.05) increase in phosphatase activity of Cap8C. All the four concentrations (0.25, 0.5, 1.0 and 2.0 mM) of Co^2+^ studied showed increased and sustained activating effects on Cap8C-catalysed hydrolysis of pNPP. The activating effect peaked in the presence of 1.0 mM Co^2+^. Ni^2+^ had a progressive activating effect on Cap8C with a steady increase in activity as the metal ion concentration was increased from 0.25 mM to 2.0 mM. At all the concentrations of Mn^2+^ studied, high activating effects on Cap8C-catalysed hydrolysis of pNPP were observed. This activating effect peaked at 1.0 mM (nine-fold increase), and there was a slight decrease in activity as the metal ion concentration was increased from 1.0 mM to 2.0 mM. Overall, 1.0 mM Co^2+^ had the highest activating effect on Cap8C activity with an approximately 17-fold increase (Figure 5).

Wzb: A similar pattern of concentration-based activation of Cap8C was observed for Wzb (Figure 5). Co^2+^ had the highest activating effect on the Wzb-catalysed hydrolysis of pNPP. This activating effect started steadily as concentration increased from 0.25 to 0.5 mM, followed by a rapid increase as Co^2+^ concentration approached 1.0 mM where the activity peaked, yielding a 27-fold increase in activity. There was no significant change in activity as Co^2+^ concentration was increased from 1.0 mM to 2.0 mM.

Ni^2+^ showed a limited activating effect on Wzb (Figure 5) with no effect seen in the presence of 0.25 and 0.5 mM of the divalent cation. An increase in the Ni^2+^ concentration to 1.0 mM and 2.0 mM led to a corresponding increase in its activating effect of Wzb with a seven-fold increase.

Mn^2+^ showed a concentration-dependent steady activation of Wzb from 0.25 mM to 1.0 mM, where it peaked with a 26-fold increase in activity. However, increasing the concentration of Mn^2+^ to 2.0 mM led to a sharp drop-in activity, giving a 10-fold enhancement over the control compared to the 26-fold increase seen with the 1.0 mM concentration (Figure 5).

### 2.4. Temperature Dependence of Cap8C and Wzb Activities

LaPointe et al. showed that the optimum temperature of Wzb was 75 °C [7]. In addition, Phyre2 analysis of the structure of Cap8c showed some structural similarity to some thermophilic proteins. Hence, we determined the effect of temperature (30–90 °C) on the activities of Cap8C and Wzb.

Cap8C: Figure 6A shows that Cap8C exhibited high activity when activated by its cofactors at 50–90 °C. In the presence of 1.0 mM Co^2+^, the activity of Cap8C increased with temperature from 93.66 mmol/min at 30 °C to 2931.61 mmol/min at 90 °C. With the Co^2+^ ion, the activity of Cap8C peaked at 90 °C with an activity of 2931.61 mmol/min, which is 31-fold higher than that at 30 °C. Similarly, the activity of Cap8C increased with temperature in the presence of 1.0 mM Ni^2+^ from 31.93 mmol/min at 30 °C to 543.33 mmol/min at 90 °C (Figure 6). However, at 70 °C, this activity peaked (609.84 mmol/min), yielding a 19-fold increase in activity when compared with the activity observed at 30 °C. In the presence of 1.0 mM Mn^2+^, Cap8C activity increased with temperature from 49.31 mmol/min at 30 °C to 503.95 mmol/min at 90 °C. Cap8C activity peaked at 90 °C in the presence of Mn^2+^ with a 10-fold increase (Figure 6).

Wzb: The effect of temperature (30–90 °C) on Wzb-catalysed hydrolysis of pNPP in the presence of the divalent cations (Co^2+^, Ni^2+^ and Mn^2+^) is shown in Figure 6B. The result revealed that in the presence of 1.0 mM Co^2+^, the activity of Wzb increased rapidly with an increase in temperature, peaking at 90 °C. With Co^2+^, Wzb activity at 90 °C (4696.49 mmol/min) was six-fold higher than that observed at 30 °C. When Ni^2+^ was used as the cofactor, the activity of Wzb also increased with temperature, peaking at 90 °C (1132.95 mmol/min), representing a four-fold increase. This increase in activity is significantly less than what was seen when Co^2+^ was used as the cofactor. In the presence of 1.0 mM Mn^2+^, Wzb activity rapidly increased with temperature and peaked at 70 °C with an activity of 3092.65 mmol/min (~3-fold increase). But as the temperature approached 90 °C, this activity declined rapidly from 3092.65 to 904.67 mmol/min (~three-fold decrease).

## 3. Discussion

### 3.1. Divalent Metal Ions as Cofactors of Cap8C and Wzb Phosphatase Activities

The AlphaFold2-predicted structures of Cap8C and Wzb put the two protein tyrosine phosphatases (PTPs) in the polymerase and histidinol phosphatase (PHP) family (Figure 1). PTPs are generally not considered to have catalytic activities that are dependent on metal ions except for those in the PHP superfamily [4,6]. However, some PTPs have been reported to have activities modulated by the presence of some metal ions [19,20,21]. Although phosphatases of the PHP superfamily are co-purified with metal ions attached to the active site, their activities in vitro in the absence of the exogenous supply of some metal ions remain barely detectable [12,13,14], indicating nonstructural roles for the metal ions. Hence, one of the objectives of this work was to identify the role of divalent metal ions as cofactors in the molecular mechanism of Cap8C and Wzb. We chose the two enzymes as representative members of the family to see if there were any subtle differences in their catalytic properties. The predicted catalytic site of PHP, which consists of four motifs with conserved histidine residues, has been proposed to be involved in the metal-dependent catalysis of phosphoester bond hydrolysis [7,14] through metal ion binding and coordination of enzymatic activity [14]. Also, the conserved aspartate residue of the fourth motif has been predicted to be important for catalysis by participating in electron transfer [9].

Previous reports have shown that Co^2+^, Ni^2+^, Mn^2+^ and Mg^2+^ stimulate phosphatase activity to different extents, while Ca^2+^ has little or no activating effect on phosphatases [5]. The effect of these selected metal ions on the phosphatase activities of Cap8C and Wzb was investigated in this study, with the aim of establishing their dependence on these cations. Our results show that in the absence of Co^2+^, Ni^2+^ and Mn^2+^, the activities of Cap8C and Wzb on pNPP hydrolysis were barely detectable at pH 7.5 (Figure 2). This shows both enzymes are metalloenzymes that require the presence of metal ions for full activation, just as reported for members of the PHP superfamily that show low activities in the absence of metal ions [8,12]. Mijakovic et al. reported a metal-dependent PTP of the PHP superfamily YwqE from *B. subtilis*, with barely noticeable activity in the absence of 1.0 mM Mn^2+^, 1.0 mM Cu^2+^ and 1.0 mM Zn^2+^ [22]. A baseline level of phosphatase activity was also observed for *S. pneumoniae* CpsB in the absence of 1.0 mM Co^2+^, 1.0 mM Mn^2+^ and 1.0 mM Mg^2+^ [9].

Our results show that Co^2+^ best activated both Cap8C and Wzb with a 17-fold and 29-fold increase in activity, respectively. This study is the first to report Co^2+^ activation of Cap8C activity and shows that the PTP behaves in a similar pattern in the presence of Co^2+^ as other members of the PHP family.

The PHP family members of PTPs have been generally described to have optimal activity at basic pH and are also sensitive to Mn^2+^ [13,23]. *Streptococcus pneumoniae* CpsB, the first PTP representative of the PHP superfamily to be described, has phosphatase activity dependent on Mn^2+^ [11]. Like *S. pneumoniae* CpsB, *B. subtilis* YwqE (another member of the PHP superfamily) exhibited maximal phosphatase activity in the presence of Mn^2+^ [22]. Similarly, Standish et al. reported that DNA polymerase PolC exonuclease activity was significantly higher in the presence of Mn^2+^ than in Mg^2+^ [23]. In our study, both Cap8C and Wzb were significantly activated by Mn^2+^, 7-fold and 24-fold, respectively. The characteristic activation of Cap8C and Wzb by Mn^2+^ reveals a significant difference from what was described for *S. pneumoniae* CpsB (manganese-dependent PTP) [9]. This suggests that although Cap8C has been proposed to be a homologue of *S. pneumoniae* CpsB, its sensitivity to Mn^2+^ differs and its phosphatase activity does not seem to be dependent on the cation. The responsiveness of Cap8C to Mn^2+^, in addition to the earlier claim of sequence homology with the novel member of the PHP superfamily, *S. pneumoniae* CpsB, shows that Cap8C belongs to the PHP family.

The conserved histidine residues on the catalytic motif of phosphatases of the PHP superfamily have been suggested to be involved in metal binding, especially of Ni^2+^ [24,25]. Although studies are yet to confirm that Ni^2+^ is essential for activating phosphatases of the PHP superfamily, it has, however, been shown to activate them to a lesser extent compared with the activating effects of Co^2+^, Cu^2+^ and Mn^2+^ [5,12]. In this study, we show that Ni^2+^ significantly increased the activities of Cap8C and Wzb (Figure 4). Ni^2+^ was observed to have a higher activating effect on Cap8C (12-fold) than on Wzb (4-fold). LaPointe showed that the absence of Ni^2+^ in reaction mixtures containing other cations resulted in a minimal reduction in the phosphatase activity of Wzb [7]. Also, the activity of *S. pneumoniae* CpsB, the pioneer phosphatase of the PHP superfamily in the presence of Ni^2+^, was observed to have increased by approximately five-fold [12]. Shi et al. reported that the phosphatases from the phosphoprotein phosphatase (PPP) superfamily’s activities were significantly activated by Ni^2+^ and they include *S. typhimurium* PrpA, *S. typhimurium* PrpB, *B. subtilis* PrpE and *S. coelicolor* SppA [26]. However, the specific role of Ni^2+^ in these systems is yet to be established.

Mg^2+^ and Ca^2+^ ions are generally not known to be essential for the activity of PTP members of the PHP superfamily but can stimulate to some extent the activity of some phosphatases [5]. Based on their ionisation potential, Mg^2+^ and Ca^2+^ are not considered to possess strong Lewis acidity when compared to other divalent metal ions such as Co^2+^, Ni^2+^ and Mn^2+^. This might contribute to their poor activating effect during the catalysis of hydrolysis reactions [27]. Our results show that Mg^2+^, Ca^2+^, Cu^2+^, Zn^2+^ and Fe^2+^ had no significant effect on the phosphatase activity of Cap8C and Wzb, as both enzymes only yielded baseline activities in the presence of the cation (Figure 4). These results agree with what has been reported for other PHP PTPs. Mijakovic et al. showed that Mg^2+^ and Ca^2+^ have no effect on the phosphatase activity of *B. subtilis* YwqE [22]. Morona et al. also reported a baseline level of phosphatase activity for *S. pneumoniae* CpsB in the presence of Ca^2+^ [9].

### 3.2. Activation of Cap8C and Wzb by Divalent Cations Is Concentration-Dependent

With the activating effects of Co^2+^, Ni^2+^ and Mn^2+^ on Cap8C and Wzb already established in the previous section, the effect of varying concentrations of these ions on the phosphatase activity of Cap8C and Wzb was investigated. The findings generally showed that the metal ions had a concentration-dependent activating effect on the phosphatase activities of both enzymes (Figure 5).

Protein tyrosine phosphatases of the PHP superfamily have been reported to be sensitive to Mn^2+^ [13], and the findings from this study show that Cap8C and Wzb are no exception. The findings in this study reveal that all concentrations of Mn^2+^ studied (0.25, 0.50, 1.00 and 2.00 mM) had a high and sustained activating effect on Cap8C. But even at its peak (1.0 mM), the activating effect of Mn^2+^ was observed to be lower than what was obtained in the presence of Co^2+^. This suggests that although Cap8C is sensitive to Mn^2+^, the enzyme’s activity was maximal in the presence of Co^2+^. On the other hand, for all the concentrations used in this study, Mn^2+^ had a progressive activating effect on Wzb that peaked at 1.0 mM (Figure 5). The findings in this study reveal that Cap8C and Wzb pose slightly different sensitivity to Mn^2+^, as the cation was observed to favour the phosphatase activity of Wzb more. This finding agrees with the reports by LaPointe et al., where 0.1 mM Mn^2+^ was reported to only increase the activity of *L. rhamnosus* Wzb by two-fold [7]. Hagelueken et al. reported that 0.5 mM Mn^2+^ resulted in a 13-fold increase in the phosphatase activity of *S. pneumoniae* Cps4B [12]; this is, however, higher than what was obtained in our study for both enzymes.

Our findings show that the activating effect of Co^2+^ on Cap8C and Wzb was optimal at 1.0 mM for both enzymes. This trend of activation for Cap8C and Wzb suggests that Co^2+^ might be performing the same role during catalysis by both enzymes, as the patterns of activation were similar for both enzymes. Also, the role of Co^2+^ in the catalysis by both enzymes may not be structural, as many metal ions that have been implicated in structural functions during catalysis are reported to be deeply buried within the active site of enzymes [28]. The metal ions that carry out structural functions act by helping the enzyme adopt proper conformation, which promotes interactions and consequently enhances enzymatic catalysis [28]. Crystal structures that have been determined for some members of the PHP family have not shown Co^2+^ to be part of the metal ion cluster at the active site of the purified enzymes [12,13]. Kim et al. modelled the metal ion cluster at the active site of YwqE from *Bacillus subtilis* and CpsB from *S. pneumoniae* as two iron (Fe) and magnesium (Mg) ions [12]. It follows that a structural role for Co^2+^ in Cap8C is a possibility that is worthy of further exploration.

Generally, Ni^2+^ has been reported to be a poor activator of phosphatases, especially of the polymerase and histidinol phosphatase superfamily [5]. For Cap8C, unlike other phosphatase members of the PHP superfamily, the activating effect was observed to be high. On the other hand, Ni^2+^ had the least activating effect on Wzb, as expected for PTP members of the PHP superfamily. Although the activating effect of Ni^2+^ was maximal at 2.0 mM, Co^2+^ was a better activator of Cap8C than Ni^2+^. The pattern of Co^2+^ and Ni^2+^ activation suggests that the cations might be influencing enzyme–substrate binding by facilitating the release of protons from bound water, yielding nucleophilic hydroxyl ions [12,28]. These cations may also boost Cap8C activity by interacting with negatively charged amino acid residues such as aspartic and glutamic acid at the active site, thereby stabilizing negative charges of trigonal–bipyramidal intermediate, thus translating into increased activity [8,9]. In agreement with our findings in this study, 0.5 mM Ni^2+^ was observed to increase the activity of wild-type Cps4B from *S. pneumoniae* by five-fold [12]. LaPointe et al. reported that the absence of Ni^2+^ in a reaction mixture that contained other metal ions resulted in a slight decrease in *L. rhamnosus* Wzb activity [7].

The summary of the investigation carried out on the effect of varying concentrations of metal ions (Co^2+^, Ni^2+^ and Mn^2+^) on the phosphatase activity of Cap8C and Wzb revealed that both enzymes had higher levels of dependence on Co^2+^ compared to the other two cations. These characteristics imply that although both Cap8C and Wzb belong to the PHP superfamily, they are significantly different from the phosphatase described for *S. pneumoniae*, CpsB.

We observe that relatively high metal ion concentrations were required to achieve optimal activities for the two PTPs (Figure 3 and Figure 4). As discussed above, one likely reason could be that the proteins require the metal ions for both catalytic and structural roles [29], hence a higher than usual amount would be required to saturate the protein cation binding site. Since we have not measured the binding affinities of the metal ions for the proteins, this remains a speculation at best. It is also likely that the polyhistidine tag used for protein purification could sequester some divalent cations, making them unavailable at the protein binding sites where they are required.

### 3.3. Temperature Dependence of Cap8C and Wzb Activities

A rise in temperature is expected to exponentially increase the rate of enzyme-catalysed reaction, at least until a point is reached where the observable decline in activity begins due to the loss of the protein’s native structure [30]. For an enzyme-catalysed reaction, an increase in temperature below the optimal temperature usually results in a corresponding increase in activity. This is partly because the inactivation effect of temperature is not pronounced at suboptimal temperatures [30]. However, at supraoptimal temperatures, activity decreases because of increased thermal-induced enzyme inactivation [30]. Thermophilic and hyperthermophilic enzymes are intrinsically active and stable at high temperatures and they offer major biotechnological advantages over mesophilic or psychrophilic enzymes. They are easily purified by heat treatment, are more resistant to chemical denaturation, have higher reaction rates and tolerate higher substrate concentrations [31,32].

The effect of temperature (30–90 °C) on the phosphatase activities of Cap8C and Wzb was investigated in the presence of Co^2+^, Ni^2+^ and Mn^2+^. This was executed to understand the catalytic behaviour and stability of these enzymes in the presence of the divalent ions at various temperatures.

The results of this study showed that in the presence of Co^2+^, Ni^2+^ and Mn^2+^, Cap8C showed a thermostable tendency as activities were observed to increase with temperature beyond what is observed for enzymes of mesophilic origin. In the presence of Co^2+^, Cap8C had optimal activity at 90 °C, which is significantly higher than the optimal growth temperature (37 °C) of *S. aureus*. This indicates that in the presence of the cation, an increase in Cap8C activity was maintained with the increase in temperature. This suggests that as stabilizing bonds of Cap8C break and reform rapidly, Co^2+^ is able to promote and maintain conformational changes in the protein needed for catalysis to occur at a faster pace [33]. The results also showed that in the presence of Co^2+^, Cap8C was more resistant to heat-induced inactivation that might have taken place at high temperatures.

Although with lower levels of activities, in the presence of Ni^2+^ and Mn^2+^, Cap8C activity increased steadily with the increase in temperature, with optimal activity occurring at 70 and 90 °C, respectively (Figure 4), compared to the 37 °C optimal growth temperature of *S. aureus*. The result implies that the binding of Ni^2+^ and Mn^2+^ to Cap8C helped to some extent in stabilizing the conformational changes necessary for catalysis to take place at high temperatures. In summary, our findings showed that in the presence of the metal ions, Cap8C displayed thermostable potentials in the order Co^2+^ > Ni^2+^ > Mn^2+^ and the thermostability is higher than what was reported for other PTPs from the same species [34]. Earlier, PTPa and PTPb from *S. aureus* were characterised and both enzymes had optimal activity at 40 °C [34]. A protein tyrosine phosphatase from mesophilic fungi, *Metarhizium anisopliae*, has been reported to show thermostable properties in the absence of metal ions, with optimum activity between 70 and 75 °C [35].

Like Cap8C, Wzb had thermophilic properties in the presence of the metal ions, although to a different extent, as its activity remarkably increased with temperature. With Co^2+^, Wzb had a progressive increase in activity with the increase in temperature, which peaked at 90 °C; this is significantly higher than the optimal growth temperature (6 to 41 °C) of *L. rhamnosus*.

The finding that Cap8C and Wzb are PTPs found in mesophilic organisms with thermophilic properties in the presence of Co^2+^, Ni^2+^ and Mn^2+^ is not unprecedented. LaPointe et al. reported the thermostable properties of *L. rhamnosus* Wzb in the presence of metal ions with activity peaking at 75 °C; however, the thermostability was studied in the presence of Cu^2+^, Co^2+^, Fe^3+^, Mn^2+^ and Mg^2+^ in combination [7]. John et al. have also characterised a thermostable protein tyrosine phosphatase from *Trypanosoma evansi* with the optimal temperature at 70 °C [36]. This thermostable PTP, as described for *T. evansi*, was suggested to play an important role in the adaptation of the organism to extreme temperatures in vivo. In addition to the claim that thermostable protein tyrosine phosphatase can be found in mesophiles is the purification and characterisation of thermostable PTP from *Metarhizium anisopliae*, which showed optimal activity at 70 °C [37]. Zhang and colleagues suggested that the high percentage of polar amino acid and proline content in *Metarhizium anisopliae* PTP might be responsible for its thermostability, as these features play significant roles in protein stability [37].

The differences observed in the effect of the three cations on Wzb and Cap8C could be due to subtle differences in the structure of the proteins or within the local environment of the catalytic residues. Such differences are not obvious in the AlphaFold-predicted structures. A careful examination of the amino acid sequence of the four conserved motifs shows some differences between Wzb and Cap8C among the non-conserved residues (Figure 1). More extensive characterisation and mutational studies of the two enzymes and other members of the PHP and PTP families will be required to provide more useful insight into the structural origins of these differences.

Bacteria can be classified based on their optimal growth temperature into four groups; psychrophiles (−5 to 15 °C), mesophiles (15 to 45 °C), thermophiles (45 to 80 °C) and hyperthermophiles (≥80 °C) [38]. Generally, enzymes that are synthesised by thermophiles and hyperthermophiles are regarded as thermozymes; they are usually thermostable or resistant to irreversible inactivation at high temperatures and are thermophilic (optimally active at temperatures between 60 and 125 °C) [32]. Psychrozymes originate from psychrophiles, while mesophiles produce enzymes known as mesozymes [39]. *S. aureus* and *L. rhamnosus* are typical examples of mesophiles with optimum growth temperatures (OGTs) of 7–48 °C and 6–41 °C, respectively [40]. *Lactobacillus rhamnosus* Wzb is important for extracellular polysaccharide production and polymerisation [41], and *Staphylococcus aureus* Cap8C is part of a group of enzymes required for the biosynthesis of Capsular polysaccharide [42]. There are reports of mesozymes with thermophilic potential. For example, two thermophilic restriction endonucleases, PtaI and PpaAII, from mesophilic cyanobacteria strains have been reported with optimum activity at 65–80 °C, which is far above the lethal temperature of the parent organism (40 °C) [43]. LaPointe et al. showed that *L. rhamnosus* Wzb had optimal activity at 75 °C, which is almost 30 °C higher than its OGT [7]. These interesting albeit unusual patterns are very good and require further investigation.

Merone et al. reported a thermostable PHP phopshotriesterase from *Sulfolobus solfataricus* that is activated by Mg^2+^, Co^2+^ and Ni^2+^ ions [44]. Further studies on this and other systems could help provide insight into the structural and mechanistic basis for the thermophilic properties of these enzymes. Baros et al. also showed that mutations in the degenerate metal-binding site of *E. coli* DNA polymerase III PHP domain decrease the overall stability of the polymerase and reduce its activity [45]. This suggests that the observed thermophilic properties of the PHP PTPs might be linked to the unique distorted (α/β)7 barrel of the PHP domain.

## 4. Materials and Methods

### 4.1. Reagents

All reagents including salts of divalent metal ions and the reaction substrate (p-nitrophenylphosphate) were purchased from Sigma-Aldrich, Gillingham, UK.

### 4.2. Cloning, Expression and Purification of Wzb and Cap8C

Modified versions of the molecular cloning protocols of Sambrook and Russell [46] were used for the cloning of plasmid vectors. Codon-optimised sequences of Lactobacillus rhamnosus Wzb (GenBank: AAW22448) and Staphylococcus aureus Cap8c (GenBank: AAB49432) were obtained from GeneArt (Invitrogen, Renfrewshire, UK). The coding sequences were cloned into the pET-28a(+) vector (Novagen, Worcestershire, UK) between the NdeI and XhoI sites for use in protein overexpression. The resulting pET-28a(+) vector carries a tag that contains an N-terminal hexahistidine tag (MGSSHHHHHHSSGLVPRGSHM) to facilitate purification via nickel affinity chromatography. Both plasmid vectors were sequenced (Eurofins Genomics, Wolverhampton, UK) to verify the sequence of the protein tyrosine phosphatase open reading frame.

Expression and purification of Wzb and Cap8C are as described in Sulyman et al. [47]. Briefly, the BL21(DE3)pLysS strain of *E. coli* was transformed with the pET-28a(+) vectors of Wzb and Cap8C and protein expression was induced with 0.5 mM IPTG. The histidine-tagged Cap8C and Wzb were purified using nickel affinity chromatography, and the selected fractions were dialysed in protein dilution buffer (PDB; 25 mM Tris–HCl (pH 7.5), 1 mM DTT, 1 M NaCl and 50% *v*/*v* glycerol) and stored at −20 °C.

### 4.3. Assay of Protein Tyrosine Phosphatase Activity

Phosphatase reactions were initiated by the addition of 5 mM pNPP to a reaction mixture containing 0.25 µM Cap8C/Wzb and 92 mM Tris–HCl, pH 7.5, in the presence or absence of metal ions. Each metal ion (Co^2+^, Mn^2+^, Ni^2+^, Mg^2+^ and Ca^2+^) was added at a final concentration of 1.0 mM. The assay was carried out at 75 °C for 15 min. The reaction was stopped by the addition of 100 µL of stop buffer (containing 25 mM Tris–HCl, pH 7.5, 25 mM EDTA and 0.1% SDS); the absorbance was read at 405 nm (characteristic for the reaction product p-nitrophenol) against the blank. The blank contained everything from the test samples, except that Cap8C/Wzb was substituted with PDB (protein dilution buffer). Absorbance readings were taken on a Clariostar Microplate Reader (BMG Labtech, Ortenberg, Germany). All measurements of reaction rate were performed in triplicate, and activities are expressed as changes in absorbance at 405 nm per minute.

In experiments where the effects of temperature were investigated, reactions were incubated at 30, 40, 50, 60, 70, 80 and 90 °C for 15 min, after which the reaction was stopped by the addition of 100 µL of stop buffer.

In pre-incubation experiments to test the effect of metal ions on the thermostability of the enzymes, a reaction mixture containing 0.25 µM Cap8C/Wzb, 92 mM Tris–HCl, pH 7.5 and metal ions (Co^2+^, Ni^2+^ or Mn^2+^) was pre-incubated at different temperature (50, 60, 70, 80 and 90 °C) for 0, 2, 5, 10, 15, 20 and 30 min. The reaction mixtures were cooled on ice then the reaction was initiated by the addition of 5 mM pNPP to the reaction mixture, and the assay was carried out at 75 °C for 15 min.

### 4.4. Data and Statistical Analysis

Data analysis was performed using Microsoft Excel 2016 and Duncan multiple range test. Each value is a mean of three determinations ± S.E.M. Differences between means were considered different for *p* < 0.05.

## 5. Conclusions

The purification and characterisation of Cap8C and Wzb are of particular importance since basic analysis of the enzyme complex system is an important step in gaining an understanding of their biological functions. Although several enzyme catalytic systems have been characterised in the past decades with most newly found enzymes falling neatly into well-known patterns, the system described here seems to represent a novel paradigm. The unusual preference for high temperatures for optimal activities of the two enzymes presents a curious structural and mechanistic challenge, the reason being that the two enzymes are from mesophilic organisms rather than thermophiles. It is common to find enzymes from thermophilic organisms that have preferences for high temperatures for their catalytic activities. Such enzymes often possess structural domains or motifs that are resistant to heat denaturation. An example is Taq polymerase from *Thermus aquaticus*; an enzyme routinely used in polymerase chain reactions (PCR), *T. aquaticus* is found in hot springs and Taq polymerase is remarkably heat-stable, both structurally and functionally [48]. Further studies would be required to understand the structural and mechanistic origin of the preference for high temperature by Cap8C and Wzb. It may be that these two enzymes represent a new family of heat-loving enzymes with no immediate obvious known structural features that account for such thermophilic properties.

Findings from this study reveal the distinct dependence of Cap8C and Wzb on metal ions and unique catalytic mechanisms that differ from their eukaryotic counterparts. These findings are significant as PTPs have been described as notoriously undruggable, meaning that it is often difficult to find specific inhibitors that are not general and wide-acting protein inactivators [37,49]. The metal ion dependence properties of Cap8C and Wzb and their mode of activation described here may provide an important starting point for the development of more efficient antibacterial drugs. In addition, the findings could serve as a foundation on which studies on the mechanism of *S. aureus* and *L. rhamnosus* extracellular polysaccharide and capsular polysaccharide biosynthesis and pathogenesis can be based.

## Figures and Tables

**Figure 1 ijms-25-01262-f001:**
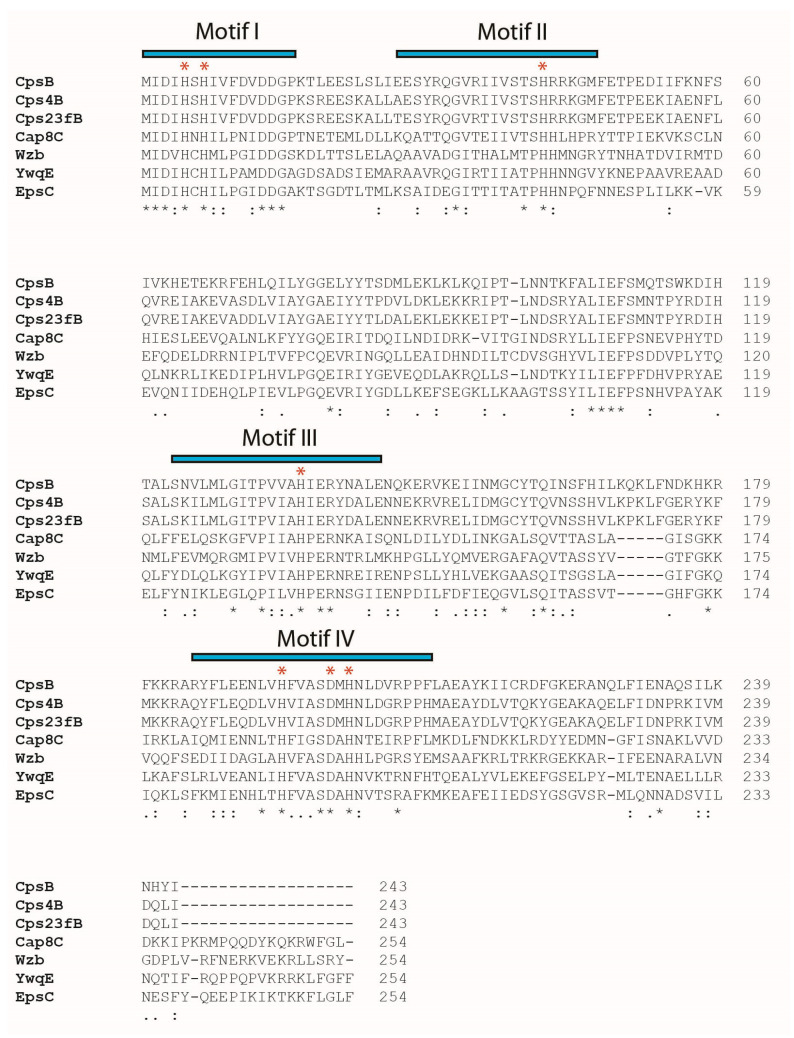
Sequence alignment of members of the PHP protein tyrosine phosphatase family including Wzb and Cap8C. The PTPs shown are *Streptococcus pneumoniae* Cps23fB (AAC69525), CpsB (AF163833), Cps4B (BEL22094); *Staphylococcus aureus* Cap8C (AAB49432); *Lactobacillus rhamnosus* Wzb (AAW22448); *Bacillus subtilis* YwqE (P96717); and *Lactococcus lactis* EpsC (NP_053031). The conserved motifs of the PHP domain [8] are highlighted as shown, while conserved amino acids involved in metal ion binding are highlighted by red asterisks. Multiple sequence alignments were generated using Clustal Omega (https://www.ebi.ac.uk/jdispatcher/msa), accessed on 4 January 2024.

**Figure 2 ijms-25-01262-f002:**
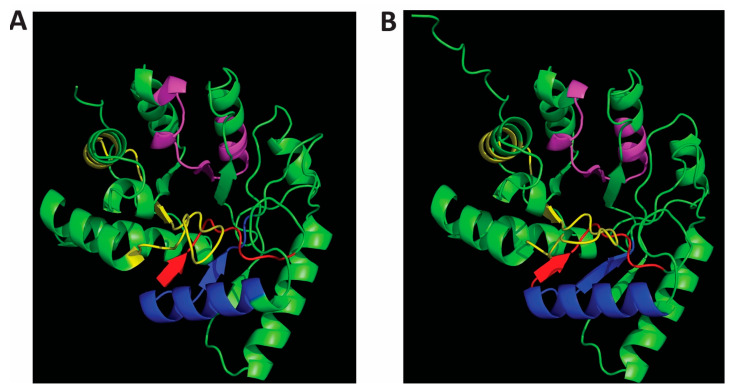
AlphaFold-predicted structures of *Staphylococcus aureus* Cap8C (**A**) and *Lactobacillus rhamnosus* Wzb (**B**). The PDB files (Wzb: AF-A0A806JCW3-F1-; Cap8C: AF-P72369-F1) were downloaded from the AlphaFold Protein Structure Database (https://alphafold.ebi.ac.uk/, accessed on 13 January 2024) and visualised with PyMOL (https://pymol.org/2/, accessed on 13 January 2024). The characteristic PHP motifs [14] are shown for both structures: Motif I (red), Motif II (blue), Motif III (magenta) and Motif IV (yellow).

**Figure 3 ijms-25-01262-f003:**
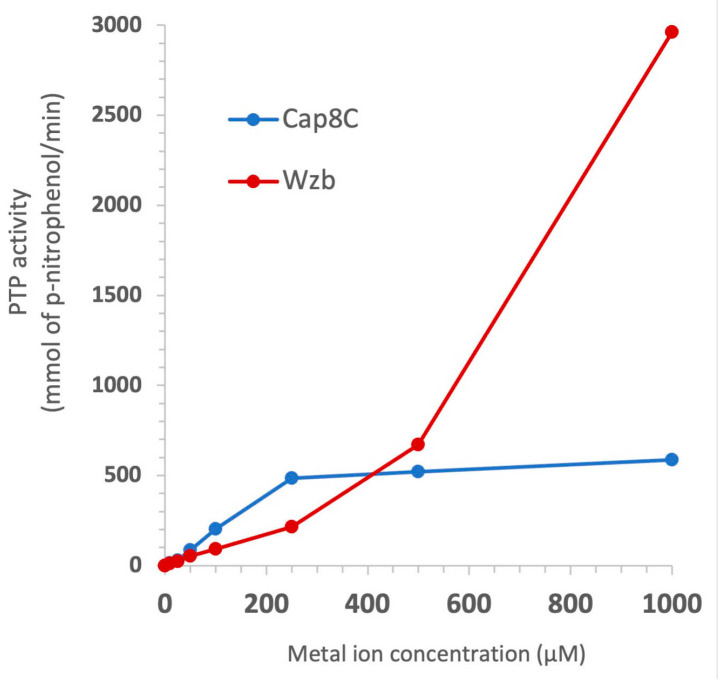
Effect of Mn^2+^ concentration on the phosphatase activities of Cap8c and Wzb. Each reaction was initiated by the addition of 5 mM pNPP to a reaction mixture containing 0.25 µM Cap8C/Wzb and 92 mM Tris–HCl pH 7.5 in the presence of indicated concentrations of Mn^2+^.

**Figure 4 ijms-25-01262-f004:**
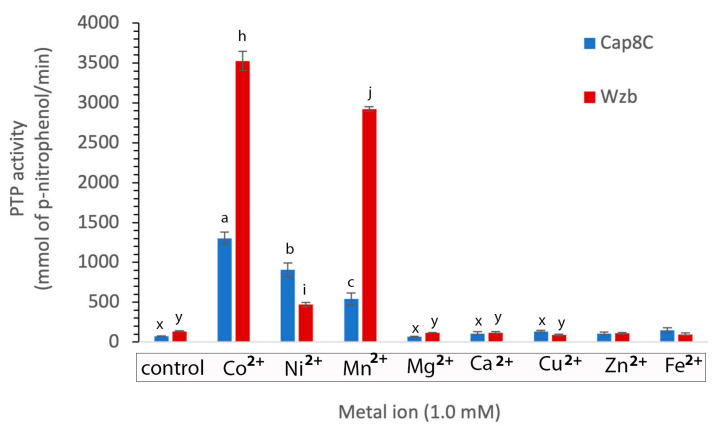
Effects of divalent metal ions on Cap8c and Wzb activities. Reactions are as described in Figure 3, and each metal ion (Co^2+^, Mn^2+^, Ni^2+^, Mg^2+^ and Ca^2+^) was added at a final concentration of 1.0 mM. The blank contained everything from the test samples, except Cap8C/Wzb was substituted with PDB (protein dilution buffer). Each value is a mean of three determinations ± S.E.M. Bars with different alphabets for each enzyme are significantly different (*p* < 0.05).

**Figure 5 ijms-25-01262-f005:**
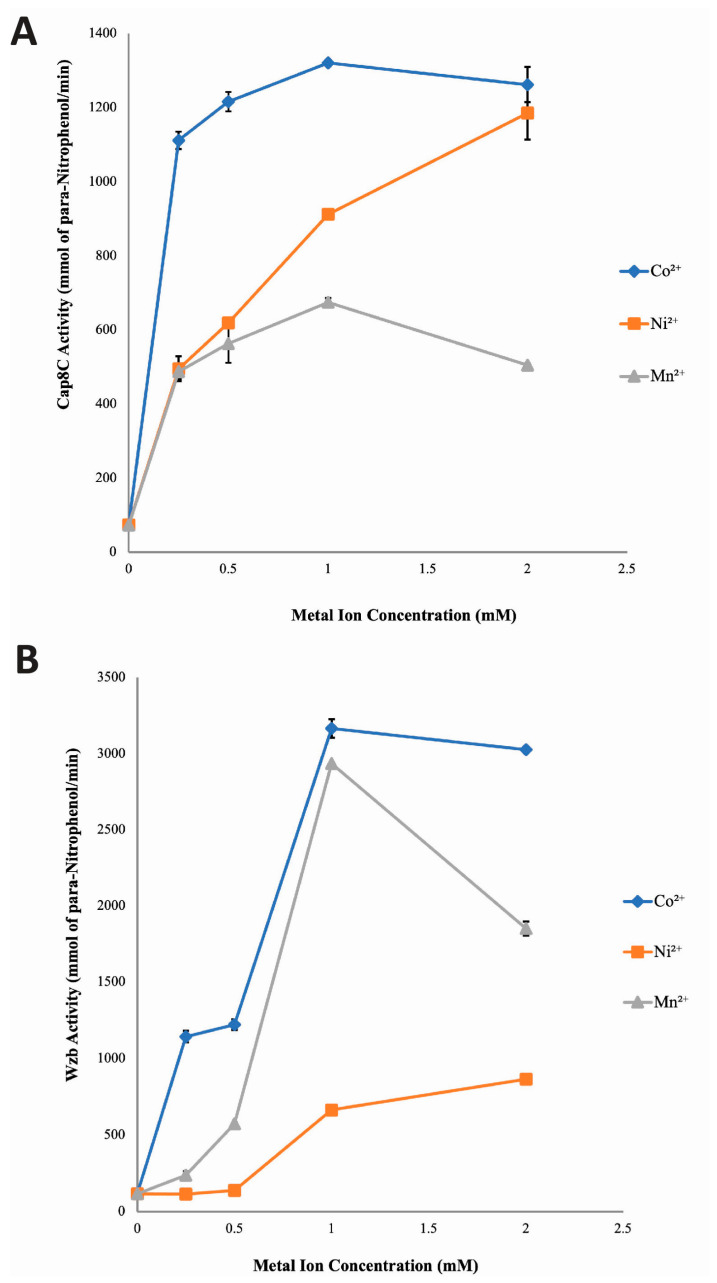
Effect of increasing concentrations of Co^2+^, Mn^2+^ and Ni^2+^ on the activities of Cap8C (**A**) and Wzb (**B**). Each metal ion (Co^2+^, Ni^2+^ and Mn^2+^) was added at a final concentration of 0.25, 0.5, 1.0 and 2.0 mM. The assay was carried out at 75 °C for 15 min and treated as described in Figure 3. Values shown at each point represent means ± SD of three replicate determinations.

**Figure 6 ijms-25-01262-f006:**
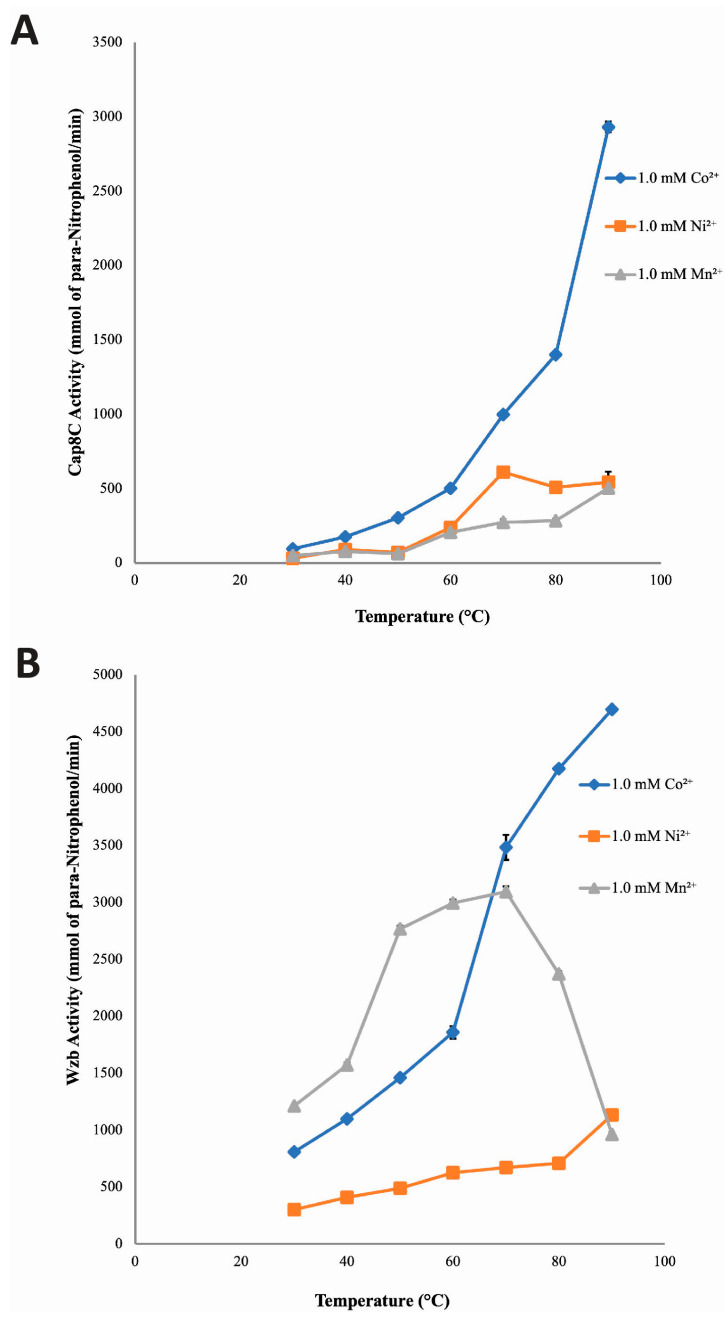
Temperature dependence of the catalytic activities of Cap8C (**A**) and Wzb (**B**) in the presence of Co^2+^, Ni^2+^ and Mn^2+^ as catalytic cofactors. Each metal ion (Co^2+^, Ni^2+^ and Mn^2+^) was added at a final concentration of 1.0 mM, and assays were carried out at different temperatures (30, 40, 50, 60, 70, 80 and 90 °C) for 15 min. All other conditions are as described in Figure 3. Values shown at each point represent means ± SD of three replicate determinations.

## Data Availability

Data is contained within the article.

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
