# Peer review of "Thermophilic PHP Protein Tyrosine Phosphatases (Cap8C and Wzb) from Mesophilic Bacteria"

_ijms, 2024, doi:10.3390/ijms25021262_

Round 1

Reviewer 1 Report

Comments and Suggestions for Authors

The work studied the enzyamtic activity of Cap8C and Wezb in the presence of metal ions as well as the  thermophilic stability. To the reviewer, the manuspt has to be greatly improved.

1) the structur of the conent has to be well organized and it is difficult to follow

2) Major points:

a.) The proteins were purified with his-tag, but His-tag is a good transition metal chelator and the results can be compromised in the method the authors used. This has to be specified.

b. ) The protein structure was predicted by AI, and the authors claimed there was metal binding site, the binding motif has to be specificed.

c.) I suggest the authors to perform ITC assay to evaluate the metal binding affinity for these two proteins.

Author Response

Referee 1:

Comment 1: The work studied the enzyamtic activity of Cap8C and Wezb in the presence of metal ions as well as the  thermophilic stability. To the reviewer, the manuspt has to be greatly improved.

1) the structur of the conent has to be well organized and it is difficult to follow

Response to Comment 1: We fully agree with the Referee that manuscript requires some major reorganisation and improvement. We have made several changes and removed some data sets that need further experimental work in order to draw clear conclusions. The data presented and the discussion of their implications are significantly clearer now.

2) Major points:

Comment 2: a.) The proteins were purified with his-tag, but His-tag is a good transition metal chelator and the results can be compromised in the method the authors used. This has to be specified.

Response to Comment 2: This is a valid point, and it has been specified in the revised version. The following statement has been added to Section 3.2. “It is also likely that the polyhistidine tag used for protein purification could sequester some divalent cations making them unavailable at the protein binding sites where they are required.”

Comment 3: b. ) The protein structure was predicted by AI, and the authors claimed there was metal binding site, the binding motif has to be specificed.

Response to Comment 3: The conserved motifs (Motiv I, II, III, and IV) are highlighted in the sequence alignment (Figure 1) included in the revised version. In addition, the motifs are shown in different colours in the AlphaFold-predicted structures (now Figure 2). To highlight this further, we have now added the following statement to the legend to Figure 2: “The characteristic PHP motifs [14] are shown for both structures: Motif I (red), Motif II (blue), Motif III (magenta), and Motif IV (yellow).”

Comment 4: c.) I suggest the authors to perform ITC assay to evaluate the metal binding affinity for these two proteins.

Response to Comment 4: We appreciate the suggestion to carry out ITC experiments to evaluate metal binding affinity. As stated in our response to Comment 1 above, we have revised the manuscript to focus on aspects of the work that we have complete data on so we can draw simple and straightforward conclusions. One of the points we have decided to leave out from this manuscript and marked for further studies and possibility of another manuscript is the issue of putative thermostability and the role the metal ions play in it. We believe that is where the data gathered from any ITC assay will be most useful and impactful.

Reviewer 2 Report

Comments and Suggestions for Authors

More than 30% of enzymes require metals for their activity. The study of metal dependence of metalloenzymes is one of the most important fields in biomolecular science. This manuscript suggests the enzymatic activity of two protein tyrosine phosphatase Cap8C and Wzb depending on the metal species. Although these are interesting research subjects, they are not adequately researched or discussed in this manuscript.

List some problems.

・About the two enzymes

It is difficult for readers to understand the relationship between two Protein Tyrosine Phosphatase Cap8C and Wzb. Furthermore, although this paper lists many previous studies of homologous enzymes, the relationships between these related enzymes and the two target enzymes were unclear. This is because no information has been clearly disclosed regarding the homology of amino acid sequences, and the conservation of active sites and amino acids involved in metal binding.

・About metal species and metal concentration

In this study, some interesting findings of two enzymes depending on the metal concentration were obtained. In Cap8C, sufficient activity was obtained at 0.25 mM metal. The authors should conduct tests at lower concentrations. This is because 0.25 mM is a large excess for 2.5 uM protein. I’d like to mention that transition metal concentrations in cell are kept low. On the other hand, Wzb could not performed sufficient activity unless about 1 mM of metal was added. Authors should discuss why Wzb requires extremely high concentrations of metals. Other metal species may be physiologically active, or a single metal species may not be sufficiently active. The authors have not tried all the previously reported metal species in PHP (e.g. iron and magnesium). In fact, the authors' cited references [8] and [17] clearly described that different metal species are observed in active sites. The authors lack insight into these matters.

・Experiments are insufficient to verify thermostability.

Line237. “2.5. Effect of metal ion cofactors on thermostability of Cap8C and Wzb”

This experiment is insufficient. The authors present experiments only using enzymes after addition of metals. This experiment alone cannot establish that the metal contributes to thermal stability. The authors should also perform the activity measurement of samples in which metals were added to preincubated metal-free enzymes. It is difficult to distinguish whether the metal contributes to thermostability or to refolding of the denatured enzymes. Alternatively, an experimental system that allows direct observation of thermal denaturation may be useful.

In addition, there are serious misconceptions about heat denaturation.

Line 583-585: It could be that lesser thermal energy would be required for Wzb to lose tightly bound Co2+, ultimately leading to unfolding thereby exposing the enzyme to chemical modifications (such as cysteine oxidation, deamidation and peptide bond hydrolysis) that eventually result in irreversible inactivation [30, 32].

Inactivation of Wzb is caused by irreversible thermal denaturation of the protein, and no results suggesting chemical modification were obtained.

・Not enough discussion in the “Discussion” section.

The authors only compared the differences in activity between Cap8C/Wzb and its homolog. In the discussion section, more in-depth discussions should be made. Differences in activity between the two enzymes and their homologs should be discussed based on amino acid sequences or three-dimensional structures. It may also be important to consider the physiological significance. Furthermore, there is no mention of the reason why such differences arise due to differences in metal types.

An example is given below.

Line 291-293: This trend suggests that Co2+ activation of the enzyme is temperature dependent and may indicate a structural transition mechanism that switches on the activity of the enzyme.

The authors suggested a structural transition mechanism by heating in Co2+. However, the discussion section does not consider structural transition at all.

### minor revision ###

There are multiple locations with duplicate content. At least the following three points can be cited (not all have been confirmed). You should avoid redundant expressions by refining your sentences.

Lines 100-102 is almost the same as Lines 75-77.

Lines 113-119 is almost the same as Lines 54-56.

Line 399 is almost the same as Lines 387-388.

Figure legends

The figure legend should contain a description of the figure. This is not the place to paste “material and method”.

Line 111 and 113

I couldn’t understand the reason why do you present Figure 6 and Figure 6a here?

Line 131

pNPP should be described as p-nitrophenyl phosphate without abbreviation.

Author Response

Referee 2:

More than 30% of enzymes require metals for their activity. The study of metal dependence of metalloenzymes is one of the most important fields in biomolecular science. This manuscript suggests the enzymatic activity of two protein tyrosine phosphatase Cap8C and Wzb depending on the metal species. Although these are interesting research subjects, they are not adequately researched or discussed in this manuscript.

List some problems.

Comment 1: About the two enzymes. It is difficult for readers to understand the relationship between two Protein Tyrosine Phosphatase Cap8C and Wzb. Furthermore, although this paper lists many previous studies of homologous enzymes, the relationships between these related enzymes and the two target enzymes were unclear. This is because no information has been clearly disclosed regarding the homology of amino acid sequences, and the conservation of active sites and amino acids involved in metal binding.

Response to Comment 1: We agree that a sequence alignment of Cap8C and Wzb along with other PHP protein tyrosine phosphatases is important, and we have included one (Figure 1) in the revised version. The conserved motifs are shown in the alignment and highlighted in the AlphaFold-predicted structures (now Figure 2). We have added a paragraph to the Introduction section explaining the rationale for studying the two enzymes in this work.

Comment 2:・About metal species and metal concentration

In this study, some interesting findings of two enzymes depending on the metal concentration were obtained. In Cap8C, sufficient activity was obtained at 0.25 mM metal. The authors should conduct tests at lower concentrations. This is because 0.25 mM is a large excess for 2.5 uM protein. I’d like to mention that transition metal concentrations in cell are kept low. On the other hand, Wzb could not performed sufficient activity unless about 1 mM of metal was added. Authors should discuss why Wzb requires extremely high concentrations of metals. Other metal species may be physiologically active, or a single metal species may not be sufficiently active. The authors have not tried all the previously reported metal species in PHP (e.g. iron and magnesium). In fact, the authors' cited references [8] and [17] clearly described that different metal species are observed in active sites. The authors lack insight into these matters.

Response to Comment 2: The Referee has raised some issues in this comment, and we have responded to the two key queries raised in the comment. (1) At the start of the project, based on published work, we assumed Mn2+ would be the default cofactor for the two enzymes. Hence, we tested the range of concentrations required to activate the enzymes. The data is now included in the manuscript as Figure 3. The results show that 1 mM is the concentration that gave optimal activity for both Wzb and Cap8C. These details have now been included in Section 2.3. We have added a paragraph to the end of Section 3.2 to discuss the likely reasons why high concentrations of metal ions are required for the in vitro experiments reported here. While it is true that “transition metal concentrations in cell are kept low”, the study carried out here are done in vitro and the manuscript has not attempted to ascribe a direct correlation to in vivo functions of the enzymes. In fact, the high temperature requirement of the enzyme observed in vitro is not likely to have in vivo relevance since the organisms are mesophilic. (2) As recommended by the Referee, we have conducted additional experiments to test other metal ions referred to in earlier studies. In addition to Mn2+, Ni2+, and Co2+ that showed significant activation, we have also tested Mg2+, Ca2+, Cu2+, Zn2+, and Fe2+. The results have been added to the earlier ones and shown in Figure 4 and commented on in Section 2.2. While we agree with the author that the comments made are valid, and we have responded to them in the revised version, we do not agree with the statement by the referee that “the authors lack insight into these matters”.

Comment 3:・Experiments are insufficient to verify thermostability.

Line237. “2.5. Effect of metal ion cofactors on thermostability of Cap8C and Wzb”

This experiment is insufficient. The authors present experiments only using enzymes after addition of metals. This experiment alone cannot establish that the metal contributes to thermal stability. The authors should also perform the activity measurement of samples in which metals were added to preincubated metal-free enzymes. It is difficult to distinguish whether the metal contributes to thermostability or to refolding of the denatured enzymes. Alternatively, an experimental system that allows direct observation of thermal denaturation may be useful.

Response to Comment 3: We agree with the Referee that the experimental data presented in the original manuscript are insufficient to establish that the enzymes are thermostable, by the strict definition of the term. The title of the paper focuses on the thermophilic properties of the enzymes since they have the best activities at high temperature. To simplify the message of the manuscript and to avoid confusion about what the data mean, we have removed the data on the effects of metal ions on recovered activities following pre-incubation at high temperatures. We hope to expand the scope of that aspect and design experiments to verify the origin of the enzymes’ thermophilic tendencies. Accordingly, the Results and the Discussion sections have been written to focus on (1) the metal ion preference for activation and (2) the observed increased activities at high temperature.  

Comment 4: In addition, there are serious misconceptions about heat denaturation.

Line 583-585: It could be that lesser thermal energy would be required for Wzb to lose tightly bound Co2+, ultimately leading to unfolding thereby exposing the enzyme to chemical modifications (such as cysteine oxidation, deamidation and peptide bond hydrolysis) that eventually result in irreversible inactivation [30, 32].

Inactivation of Wzb is caused by irreversible thermal denaturation of the protein, and no results suggesting chemical modification were obtained.

Response to Comment 4: As explained in the response to comment 5 above, the sections on thermal inactivation have been removed from the revised manuscript.

Comment 5:・Not enough discussion in the “Discussion” section.

The authors only compared the differences in activity between Cap8C/Wzb and its homolog. In the discussion section, more in-depth discussions should be made. Differences in activity between the two enzymes and their homologs should be discussed based on amino acid sequences or three-dimensional structures. It may also be important to consider the physiological significance. Furthermore, there is no mention of the reason why such differences arise due to differences in metal types.

An example is given below.

Line 291-293: This trend suggests that Co2+ activation of the enzyme is temperature dependent and may indicate a structural transition mechanism that switches on the activity of the enzyme.

The authors suggested a structural transition mechanism by heating in Co2+. However, the discussion section does not consider structural transition at all.

Response to Comment 5: We have added a paragraph each to Section 3.2 and Section 3.3 to discuss some of the points raised by the Referee. Since we have removed the data on the effect of metal ions on recovered activities after preheating the enzymes, hence there is no need for further discussion on those points. Overall, we are mindful of the risk of drawing conclusions that are not supported by data. Hence, we have taken a more cautious approach in assigning mechanistic or physiological significance to our findings. We believe that more work still needs to be done on these group of enzymes and we do not want to speculate too much ahead of collecting data that can provide accurate insights into how the system works. We are actively planning more detailed experiments that will allow us to address these mechanistic questions. For now, we have limited the Discussion to points that we are supported by the data shown in the manuscript.

### minor revision ###

Comment 6: There are multiple locations with duplicate content. At least the following three points can be cited (not all have been confirmed). You should avoid redundant expressions by refining your sentences.

Lines 100-102 is almost the same as Lines 75-77.

Lines 113-119 is almost the same as Lines 54-56.

Line 399 is almost the same as Lines 387-388.

Response to Comment 6: We thank the Referee for these careful observations, and we have harmonised the sections and their contents.

Comment 7: Figure legends

The figure legend should contain a description of the figure. This is not the place to paste “material and method”.

Response to Comment 7: The figure legends have been revised to remove any unnecessary details. In the revised manuscript, any information included in the figure legend are needed to provide information on how the reactions were carried out. The information in “materials and method” are generic and specific changes to such methods unique to each experiment should be reflected in the figure legend. This is standard practice to provide full information on what was done allowing full reproducibility of experimental work.

Comment 8: Line 111 and 113

I couldn’t understand the reason why do you present Figure 6 and Figure 6a here?

Response to Comment 8: We thank the Referee for spotting this mistake. It was an error in siting the figures, and it has been corrected now.

Comment 9: Line 131

pNPP should be described as p-nitrophenyl phosphate without abbreviation.

Response to Comment 9: We have corrected this.

Round 2

Reviewer 1 Report

Comments and Suggestions for Authors

The manuscript was greatly improved in the revision and I have no further concerns.

One minor comment: 1) the font sizes in the figures differ greatly; 2) in figure 4, the labeling of metal ion is not correct

Overall, the presentation has to be professional for the whole figures both in the legend and images.

Author Response

The manuscript was greatly improved in the revision and I have no further concerns.

Response: We thank the reviewer for taking the time to read and comment on the revised manuscript.

Comment 1: One minor comment: 1) the font sizes in the figures differ greatly; 2) in figure 4, the labeling of metal ion is not correct.

Response: We have corrected the labelling of metal ions in Figure 4, and have harmonised the font sizes. We are also aware that font sizes and other formatting issues are usually addressed prior to publication if the manuscript is accepted. We will make any needed adjustments to conform with the fine details of the journal’s format and any editorial requirements when the paper is being processed at the proof stage.

Comment 2: Overall, the presentation has to be professional for the whole figures both in the legend and images.

Response: We have tried to make the figures professional while preserving the integrity of the data.

Reviewer 2 Report

Comments and Suggestions for Authors

I have reviewed the revised manuscript and appreciate the effort the authors have put into addressing my previous comments. Overall, the significant revisions have notably improved the paper's clarity.

I agree with the authors on the significance of exploring enzyme activity and thermal stability under varying metal concentration conditions, distinct from the physiological environment. I also commend the authors for their cautious approach in refraining from attributing mechanistic or physiological significance prematurely in the Discussion section. For the current paper, I do not propose any additional discussions, as I understand this may introduce inaccuracies, as the authors rightly point out. However, I suggest a few minor revisions to enhance the manuscript further.

Comment 1: Newly Added Figure 1

In the alignment presented, the authors compare five enzymes, including Cap8C and Wzb. However, Cps23fB and EpsC, mentioned among the five enzymes, do not appear in the manuscript. Additionally, why not consider CpsB and Cps4B from Streptococcus pneumoniae, as cited in the literature and mentioned in the manuscript, for comparison? Is EpsC from Lactococcus lactis the same as "Lactococcus lactis HPP" from Line 111? To improve clarity, inserting markers at the amino acids involved in metal binding, as described by the authors (conserved histidine and aspartic acid residues), would enhance the reader's understanding.

Line 279: Objective of the Work

The sentence stating, "Hence, one of the objectives of this work was to identify the divalent metal ions that serve as activating cofactors for Cap8C and Wzb enzymes," may be misleading to readers, including myself. Considering the authors' clarification that the study should be regarded separately from physiological conditions, I suggest either deleting this sentence or rewording it to explicitly state that the conditions studied are not representative of physiological conditions.

I acknowledge the interest in exploring metalloenzyme activity under different physiological conditions and do not intend to diminish the value of this study.

Additional Comment for Future Research: Cobalt ions

I would like to propose a potential avenue for your future research. Given that cobalt ions exhibit absorption in visible light with varying peaks based on the coordination environment, simple titration experiments could provide insights into the binding affinity and stoichiometry of cobalt to proteins. Considering this, I encourage the authors to explore the feasibility of such experiments.

Author Response

I have reviewed the revised manuscript and appreciate the effort the authors have put into addressing my previous comments. Overall, the significant revisions have notably improved the paper's clarity.

I agree with the authors on the significance of exploring enzyme activity and thermal stability under varying metal concentration conditions, distinct from the physiological environment. I also commend the authors for their cautious approach in refraining from attributing mechanistic or physiological significance prematurely in the Discussion section. For the current paper, I do not propose any additional discussions, as I understand this may introduce inaccuracies, as the authors rightly point out. However, I suggest a few minor revisions to enhance the manuscript further.

Response: We thank the reviewer for taking the time to read and comment on the revised manuscript.

Comment 1: Newly Added Figure 1

In the alignment presented, the authors compare five enzymes, including Cap8C and Wzb. However, Cps23fB and EpsC, mentioned among the five enzymes, do not appear in the manuscript. Additionally, why not consider CpsB and Cps4B from Streptococcus pneumoniae, as cited in the literature and mentioned in the manuscript, for comparison? Is EpsC from Lactococcus lactis the same as "Lactococcus lactis HPP" from Line 111? To improve clarity, inserting markers at the amino acids involved in metal binding, as described by the authors (conserved histidine and aspartic acid residues), would enhance the reader's understanding.

Response: We have included CpsB and Cps4B in the alignment as suggested by the review. We agree that this is logical since those two proteins are referred to in the manuscript. EpsC is not the same as Lactococcus HPP in Line 111. There are several proteins of this family in the database, and we do not intend to publish a comprehensive alignment of all of them. We have chosen representatives to highlight the main findings of the work reported in this manuscript. As suggested, we have inserted markers at the conserved residues implicated in metal binding in the sequence alignment.

Comment 2: Line 279: Objective of the Work

The sentence stating, "Hence, one of the objectives of this work was to identify the divalent metal ions that serve as activating cofactors for Cap8C and Wzb enzymes," may be misleading to readers, including myself. Considering the authors' clarification that the study should be regarded separately from physiological conditions, I suggest either deleting this sentence or rewording it to explicitly state that the conditions studied are not representative of physiological conditions.

I acknowledge the interest in exploring metalloenzyme activity under different physiological conditions and do not intend to diminish the value of this study.

Response: We have recorded this statement as follows:  “one of the objectives of this work was to identify the role of divalent metal ions as cofactors in the molecular mechanism of Cap8C and Wzb”. We hope this clarifies the direction of the work as being mechanistic in nature, and not aimed at identifying their biological role beyond what is already in the published literature.

Additional Comment for Future Research: Cobalt ions

I would like to propose a potential avenue for your future research. Given that cobalt ions exhibit absorption in visible light with varying peaks based on the coordination environment, simple titration experiments could provide insights into the binding affinity and stoichiometry of cobalt to proteins. Considering this, I encourage the authors to explore the feasibility of such experiments.

Response: We appreciate the reviewer’s suggestion on this. We will see how this fits with future experimental work on theses enzymes.